# Pirtobrutinib in Chronic Lymphocytic Leukemia: Navigating Resistance and the Personalisation of BTK-Targeted Therapy

**DOI:** 10.3390/cancers17182974

**Published:** 2025-09-11

**Authors:** Stefano Molica, David Allsup

**Affiliations:** 1Department Hematology, Hull University Teaching Hospitals NHS Trust, Hull HU3 2JZ, UK; david.allsup@hyms.ac.uk; 2Centre for Biomedicine, Hull York Medical School, Hull YO10 5DD, UK

**Keywords:** chronic lymphocytic leukemia, pirtobrutinib, non-covalent BTK inhibitor, BTK inhibitor resistance, pirtobrutinib-safety

## Abstract

Whilst covalent Bruton’s tyrosine kinase (BTK) inhibitor therapy improves outcomes in chronic lymphocytic leukemia (CLL), resistance and adverse effects limit their long-term use. Noncovalent BTK inhibitors such as pirtobrutinib target effectively resistant mutations and show promising efficacy with favorable safety profiles. Molecular factors, such as BTK mutation status, influence response durability, underscoring the importance of personalized treatment. Ongoing trials aim to position pirtobrutinib within frontline regimens, potentially enhancing long-term disease management. In relapsed/refractory settings, integrating noncovalent BTK inhibitors into individualized treatment strategies—including as bridging therapies to cellular modalities like Chimeric Antigen Receptor T-cell (CAR-T)—may optimize outcomes and advance toward curative interventions in CLL.

## 1. Introduction

The introduction of the covalent Bruton’s tyrosine kinase (BTK) inhibitors ibrutinib, acalabrutinib, and zanubrutinib has revolutionized the treatment paradigm and improved outcomes for patients with advanced chronic lymphocytic leukemia (CLL) [1,2,3,4,5,6,7,8,9]. These orally administered agents, delivered on a daily basis, have demonstrated substantial clinical efficacy, yielding high response rates and enhancing patients’ quality of life [1,2,3,4,5,6,7,8,9]. When utilized as first-line therapy, BTK inhibitors can induce durable remissions extending beyond four years in approximately 70–80% of patients [10]. Despite these advances, a considerable proportion of patients discontinue therapy due to adverse events (AEs), predominantly attributable to off-target kinase inhibition [11]. The most common reasons for discontinuation include cardiac arrhythmia, most notably atrial fibrillation and pneumonia [12,13]. Additionally, between 13% and 37% of patients cease therapy due to limited efficacy or the development of resistance mechanisms [14]. Notably, discontinuation due to toxicity appears more prevalent with ibrutinib, whereas resistance-related discontinuation is observed across all available covalent BTK inhibitors [15,16].

All approved covalent BTK inhibitors irreversibly bind to cysteine residue 481 (Cys481) within the ATP-binding pocket of BTK, resulting in permanent enzyme inhibition [17,18,19]. This covalent interaction necessitates the formation of a stable, irreversible bond, which can only be reversed through de novo synthesis of BTK [20]. Prolonged selective pressure exerted by continuous inhibition fosters the emergence of somatic mutations within the BTK gene, often at the Cys481 site, that impair drug binding affinity. The most frequently observed mutations involve amino acid substitutions at Cys481, such as C481S or C481R, that reduce covalent binding stability, thereby diminishing drug efficacy [20]. Less commonly, gain-of-function mutations in downstream signaling molecules like phospholipase C gamma 2 (PLCγ2) can occur which enable B-cell receptor (BCR) signaling to persist, and can synergize with BTK mutations, thus contributing to resistance [21,22,23].

Noncovalent BTK inhibitors, such as pirtobrutinib, do not rely on covalent modification of the cysteine residue Cys481 in BTK [24]. Instead, these inhibitors reversibly associate with the ATP-binding site through a composite of noncovalent interactions, including hydrogen bonds, ionic interactions, and hydrophobic contacts. This reversible binding mode permits sustained inhibition of BTK activity even in the presence of Cys481 mutations that confer resistance to covalent inhibitors [25,26].

Clinical and preclinical studies have demonstrated that pirtobrutinib retains activity in patients harboring Cys481 mutations; however, resistance can also develop through other somatic alterations within the BTK gene that affect the binding interface [27,28]. Such genetic alterations may diminish the efficacy of both covalent and noncovalent BTK inhibitors, underscoring the complexity of resistance mechanisms and the need for continuous monitoring and development of novel therapeutic strategies [27,28].

This review provides a synthesis of clinical evidence up to July 2025, derived from a PubMed-based literature search, outlining the studies that have informed the incorporation of pirtobrutinib into the current therapeutic algorithm for CLL. By summarizing pivotal trials, dose–response findings, and safety signals, the review delineates how noncovalent BTK inhibition with pirtobrutinib has influenced treatment sequencing and patient management in CLL

## 2. Pirtobrutinib Mechanism-of-Action and Pharmacology

Unlike covalent BTK inhibitors, pirtobrutinib exhibits activity independent of C481 mutation status, including mutations such as C481S and C481R that confer resistance to earlier-generation agents [26]. This distinctive pharmacological characteristic positions pirtobrutinib as a versatile therapeutic agent for treating a spectrum of B-cell malignancies, notably Waldenström’s macroglobulinemia, diffuse large B-cell lymphoma, and CLL [25].

Structurally, pirtobrutinib interacts within the ATP-binding pocket of BTK at a site spatially distinct from the C481 residue (Figure 1A). This binding mode enables it to maintain inhibitory activity even when mutations impair covalent bond formation at C481 [24]. Biochemical assays, complemented by cell-based experimental models, have demonstrated that pirtobrutinib maintains comparable potency against wild-type BTK and mutant variants, including C481S and C481R, effectively inhibiting kinase activity regardless of mutation status. In vitro studies utilizing MEC-1 cell lines engineered to overexpress either wild-type or mutant BTK further confirm that pirtobrutinib effectively inhibits BTK autophosphorylation at tyrosine 223 (Y223) and downstream signaling pathways, independently of mutation presence [29].

Data obtained from the ongoing phase I/II trial NCT03740529 support these preclinical findings. Early treatment results indicate significant reductions in plasma levels of chemokines CCL3 and CCL4, which are biomarkers associated with disease activity in BTK mutated CLL which highlights the compound’s effective in vivo activity. Mechanistically, pirtobrutinib is a potent inhibitor of BTK autophosphorylation at Y223 (Figure 1B), a critical activation site. Although primarily classified as a back-pocket inhibitor, evidence suggests pirtobrutinib may also suppress upstream phosphorylation at tyrosine 551 (Y551), potentially through stabilization of BTK in an inactive, closed conformation. Structural insights further suggest that phosphorylation at Y551 is vital for the recruitment of hematopoietic cell kinase (HCK) to kinase-dead BTK mutants, which may contribute to residual signaling despite kinase domain inactivation, providing a rationale for the continued efficacy of pirtobrutinib [29].

Selectivity profiling reveals that pirtobrutinib exhibits extraordinary specificity for BTK, being over 100-fold more selective for BTK than for a broad panel of other kinases tested. Enzymatic profiling indicates that pirtobrutinib interacts minimally with over 98% of human kinases, a feature that likely contributes to a favorable safety profile by reducing off-target adverse effects, such as cardiotoxicity, which are associated with less selective covalent inhibitors [27].

Pharmacokinetic analyses have demonstrated that pirtobrutinib exhibits linear kinetics over a dose range from 25 mg to 300 mg daily, with an estimated half-life of approximately 20 h (Figure 1C). No dose-limiting toxicities have been observed at these doses, supporting the selection of 200 mg daily as the recommended phase II dose. This dosing achieves plasma trough concentrations capable of inhibiting approximately 96% of BTK activity, effectively ensuring continuous suppression of kinase activity throughout the dosing interval [30]. The extended half-life provides a pharmacokinetic advantage over irreversible inhibitors by maintaining sustained drug levels, thereby ensuring ongoing inhibition of newly synthesized BTK molecules and reducing the potential for disease progression due to incomplete kinase suppression [27,30].

## 3. Early Clinical Studies

Initial FDA approval for pirtobrutinib was based on the results of the multicenter phase 1/2 BRUIN study (NCT03740529), which enrolled patients with pretreated B-cell malignancies, including CLL, MCL, Waldenström macroglobulinemia, and follicular lymphoma [30]. A total of 323 patients received pirtobrutinib (LOXO-305) across seven dose levels—25 mg, 50 mg, 100 mg, 150 mg, 200 mg, 250 mg, and 300 mg administered once daily—demonstrating a linear relationship between dose and systemic drug exposure. No dose-limiting toxicities were identified, and the maximum tolerated dose (MTD) was not reached within the dose range studied. Based on safety and pharmacokinetic data, the recommended phase 2 dose was established at 200 mg daily. The most frequently reported adverse events, occurring in at least 10% of participants, included fatigue (20%), diarrhea (17%), and contusions (13%). Grade 3 or higher adverse events most commonly involved neutropenia, observed in 10% of patients. Notably, there were no reports of grade 3 atrial fibrillation or flutter; also grade 3 hemorrhage was rare, occurring in a single patient following mechanical trauma. Treatment discontinuation due to adverse events was necessary in approximately 1% of patients [30].

Efficacy assessments in 121 patients with CLL or small lymphocytic lymphoma (SLL), many of whom had previously received covalent BTK inhibitors (with a median of four prior lines of therapy), revealed an overall response rate (ORR) of 62%. Subgroup analyses indicated comparable response rates among patients with prior resistance to covalent BTK inhibitors (67%), those intolerant to such therapies (52%), patients harboring BTK C481 mutations (71%), and those with wild-type BTK (66%). At the time of analysis, the majority of responders—117 out of 121—remained progression-free, underscoring the potential durability of responses in this patient population [30].

The authors identify, however, some limitations in this pivotal investigation. Because some B-cell malignancies—most notably CLL—possess a long natural history, longer follow-up is required to more accurately evaluate the durability of responses to pirtobrutinib (LOXO-305). In addition, although initial safety signals for LOXO-305 appear favorable, extended observation will be necessary to define the full safety profile of the agent under conditions of chronic administration [30].

An updated analysis of this trial, including 317 with relapsed or refractory CLL or SLL, has indicated potential differences in treatment responses among patients with distinct genetic mutations [31]. Notably, among patients harboring mutated PLCγ2, the ORR was 56%, which is lower than the response observed in patients with BTK C481 mutations. There exists substantial heterogeneity among PLCγ2 mutations observed in patients [21,22,23]. Different mutations can vary in their functional impact on PLCγ2 activity, subcellular localization, and interaction with upstream (BTK, SYK) and downstream signaling components [21]. Consequently, not all PLCγ2 mutations yield identical phenotypic effects or drug responses; some variants may confer stronger resistance, while others produce modest or context-dependent changes. This heterogeneity implies that PLCγ2 status cannot be treated as a single binary biomarker for resistance [21,23].

Overall, the median progression-free survival (PFS) was 19.6 months. For patients who had previously received both a BTK inhibitor and a BCL2 inhibitor, the median PFS was 16.8 months. In contrast, patients who had been treated with a BTK inhibitor alone experienced a longer median PFS of 22.1 months.

Finally, pirtobrutinib was effective also in patients who had undergone all five available therapies for CLL or SLL—including BTK, BCL2, and PI3K inhibitors, as well as anti-CD20 monoclonal antibodies and chemotherapy. In these patients, the median progression-free survival was 13.8 months [31].

In this study, certain subgroups defined by prior therapy or molecular characteristics included only a small number of patients, resulting in wide confidence intervals for the primary efficacy endpoints. Nevertheless, the findings establish a basis for a phase III comparative trial of pirtobrutinib in CLL [31].

A retrospective, multi-center phase I/II analysis of the BRUIN study evaluated the safety and efficacy of pirtobrutinib monotherapy in patients who previously discontinued a BTK inhibitor due to intolerance [32]. This cohort included 78 patients, of whom 61.4% (n = 48) had CLL. The majority of participants had prior treatment with the first-generation covalent BTKi, ibrutinib. The adverse events (AEs) leading to discontinuation of prior BTKi therapy aligned with known toxicities associated with intolerance to ibrutinib, acalabrutinib, and zanubrutinib. Cardiac-related issues, particularly atrial fibrillation, were the most common reasons for stopping BTKi therapy, with 40 patients (31.5%) experiencing cardiac disorders and 30 (23.6%) specifically reporting atrial fibrillation [32].

The median follow-up duration was 17.4 months, during which patients remained on pirtobrutinib for a median of 15.3 months. The primary reasons for discontinuing pirtobrutinib were, respectively, disease progression (26.8%), AEs (10.2%), and death (5.5%). The most frequently reported treatment-emergent AEs were fatigue (39.4%) and neutropenia (37.0%). Notably, among patients who previously discontinued a BTK inhibitor due to cardiac concerns, 75% did not experience a recurrence of their cardiac AE. Importantly, no patient stopped pirtobrutinib because of the same AE that led to prior BTKi discontinuation, suggesting that pirtobrutinib may be a viable option for patients intolerant to covalent BTK inhibitors. Finally, the median PFS for the CLL/SLL subgroup was 28.4 months [32].

These findings establish that the third-generation BTK inhibitor pirtobrutinib possesses a proven efficacy and improved safety profile, with the promise to address several unmet needs associated with covalent BTK inhibitors. Furthermore, clinicians treating patients with CLL may have access to an agent that allows for the full utilization of BTK inhibition before transitioning patients to alternative therapy classes [33,34].

## 4. The Phase 3 Trial BRUIN CLL-321

Real-world data from sources such as the Flatiron Health EHR-derived database and Optum Clinformatics Data Mart suggest that after covalent BTKi (cBTKi) discontinuation, median time to treatment discontinuation ranges from 6 to 9 months across databases [35].

No prospective, randomized studies currently evaluate treatment options for relapsed/refractory (R/R) CLL following prior cBTKi therapy. For this patient subset, the applicability of the phase III MURANO trial which demonstrated improved outcomes with venetoclax plus rituximab is limited, as only about 3% (n = 5/194) had prior BCR inhibitor exposure [36]. As the population of CLL/SLL patients with prior cBTKi therapy increases, there is a growing need for effective treatments in this subgroup [11]. The BRUIN CLL-321 phase III randomized study, comparing pirtobrutinib to investigator’s choice consisting of Idelalisib plus rituximab (IdelaR) or bendamustine plus rituximab (BR), is unique as it exclusively enrolled in a prospective, randomized study patients previously treated with cBTKi [37]. In this study, prior therapy with venetoclax was permitted.

This study enrolled 238 patients, previously exposed to covalent BTKis who were randomly assigned to either receive pirtobrutinib (n = 119) or investigator’s choice (IC) (n = 119; including IdelaR [n = 82] and BR [n = 37]). The hazard ratio (HR) for PFS was 0.54 (95% CI, 0.39 to 0.75; *p* = 0.0002), with median PFS of 14 months (95% CI, 11.2 to 16.6) for the pirtobrutinib group compared to 8.7 months for the IC group. The unadjusted overall survival (OS) HR was 1.09 (*p* = 0.7202), with 18-month OS rates of 73.4% in the pirtobrutinib arm versus 70.8% in the IC arm. The median time to next treatment (TTNT) was 24 months with pirtobrutinib, compared to 10.9 months with IC (*p* < 0.0001). After a median follow-up of 17.2 months, grade ≥ 3 treatment-emergent AEs were observed less frequently in the pirtobrutinib group (57.7%) than in the IC group (73.4%). Treatment discontinuation due to AEs occurred in 20 patients (17.2%) on pirtobrutinib and 38 patients (34.9%) on IC [37].

These results are particularly relevant given that the study enrolled a heavily pretreated population: patients had received a median of four prior lines of therapy, with many having prior exposure to covalent BTK inhibitors, venetoclax, and chemotherapy—with or without anti-CD20 antibody treatments—highlighting the challenging nature of this cohort [37]. Of note, treatment benefits with pirtobrutinib were consistent across important subgroups, including those with high-risk features such as del(17p)/TP53 mutations, complex karyotype, and unmutated IGHV [37].

Data derived from the BRUIN CLL-321 clinical trial indicate that pirtobrutinib achieves a median TTNT of approximately 2.5 years in patients who are naïve to venetoclax therapy, and approximately 1.7 years in patients with prior exposure to venetoclax [37]. TTNT is a critically important endpoint as it provides a patient-centered measure of therapeutic durability and clinical benefit [38,39]. These findings imply that sequencing pirtobrutinib following venetoclax treatment may constitute a viable therapeutic approach, especially considering the observed differences in TTNT based on prior venetoclax exposure [38]. A retrospective analysis conducted by Thompson et al. suggests that venetoclax maintains its efficacy after treatment with non-covalent BTK inhibitors [40]. However, it is important to note that prospective studies evaluating this sequence are currently limited, and further clinical investigation is warranted to confirm these observations and optimize sequencing strategies in CLL management. Of note, the IdelaR/BR combination was selected as the comparator in the BRUIN CLL-321 trial because it represented a reasonable historical standard of care for relapsed/refractory CLL, particularly in patients previously treated with BTKIs. However, neither IdelaR nor BR is considered an optimal modern therapy. Their inclusion highlights these limitations, thereby underscoring the significant clinical benefit and improved safety profile of pirtobrutinib [40].

Finally, the BRUIN CLL-321 study confirmed that pirtobrutinib possesses a safety profile consistent with earlier phase 1/2 studies [30,31]. Discontinuation due to AEs was low (5.2%), and the incidences of atrial fibrillation, hypertension, and major bleeding were infrequent [40]. Notably, patients with a history of atrial fibrillation/flutter did not exhibit an increased risk of cardiac adverse events. Most bleeding episodes were mild in severity, and no cases of Richter transformation were observed during treatment with pirtobrutinib [37].

## 5. Pirtobrutinib in Patients in Richter Transformation (RT)

Preliminary results showed that pirtobrutinib demonstrated promising efficacy and a favorable safety profile in heavily pretreated patients with Richter transformation (RT). Specifically, 6 out of 9 patients enrolled in the pivotal phase I/II trial achieved at least a partial response, despite all having prior exposure to covalent BTK inhibitors [31]. The Phase II BRUIN study included a subgroup consisting of 82 patients diagnosed with Richter transformation. At baseline, the median age within this subgroup was 67 years, with approximately two-thirds of the patients being male, reflecting the demographic characteristics typically observed in this patient population [41]. A substantial proportion, approximately 90% had previously received at least one systemic therapy targeting RT. Moreover, 74% had been treated with a cBTKi, indicating a heavily pretreated cohort with limited therapeutic options. The study findings demonstrated that pirtobrutinib elicited an ORR of 50%, with a complete remission (CR) rate of 13%. Among responders, eight patients continued on therapy and subsequently underwent stem cell transplantation, highlighting the role of pirtobrutinib as a bridge to potentially curative strategies [41].

The median PFS was 3.7 months, while the median OS was 12.5 months. The 2-year OS rate was 33.5%, a notable outcome given the historically poor prognosis associated with RT. Importantly, for a disease with limited treatment options, pirtobrutinib demonstrated single-agent activity and served as a bridge to potentially curative therapies. These results support ongoing investigations of pirtobrutinib, both as monotherapy and in combination with other agents, as a promising therapeutic strategy for patients with RT [41].

Additionally, in many patients with CLL, who are typically older and burdened with substantial comorbidities, RT may be amenable to palliative management; in this context, single-agent pirtobrutinib could constitute a reasonable therapeutic option [42].

## 6. Pirtobrutinib in Fixed-Duration Regimens

While fixed-duration regimens combining cBTKi with venetoclax are well established in the upfront treatment of CLL, their application in relapsed/refractory (R/R) settings remains limited [43]. This consideration is particularly relevant, as many patients receive covalent BTK inhibitors as continuous frontline therapy—either as monotherapy or in combination with anti-CD20 monoclonal antibodies—thereby limiting the applicability of BTK inhibitor-based combination strategies in the relapsed setting [44,45].

In a phase 1b study, pirtobrutinib demonstrated notable efficacy when combined with venetoclax, with or without rituximab, in patients with R/R CLL. The overall ORR was 96%, with 40% of patients achieving CR. The 24-month PFS was 79.5%. Importantly, these responses were observed in a heavily pretreated population, with 68% having prior exposure to covalent BTK inhibitors, of whom 71% exhibited resistance to these agents. Additionally, the rate of undetectable measurable residual disease (uMRD) with a sensitivity of 10^−4^ in peripheral blood after twelve cycles was 70.8%. Early discontinuation due to disease progression was rare, occurring in only two patients. Pharmacokinetic analyses indicated no significant drug–drug interactions between pirtobrutinib and venetoclax, with exposure levels comparable to monotherapy [46].

These results further indicate that fixed-duration pirtobrutinib and venetoclax regimens are well tolerated and demonstrate sufficiently promising efficacy to warrant further investigation in this patient population [46].

Ongoing and planned studies include BRUIN CLL-322 (NCT04965493), a phase 3 trial comparing fixed-duration pirtobrutinib–venetoclax–rituximab (PVR) with venetoclax–rituximab (VR) in patients with relapsed/refractory CLL who have progressed after covalent BTKi therapy. This study represents the first large, time-limited venetoclax-based combination in a post-covalent BTKi setting, with approximately 80% of enrolled patients expected to be BTKi pretreated. Additional pirtobrutinib-based combinations are under evaluation, including the time-limited triplet of pirtobrutinib, venetoclax, and obinutuzumab (PVO) for treatment-naïve CLL or Richter transformation (NCT05536349, NCT05677919). Furthermore, the investigator-initiated phase 3 CLL-18 trial (NCT06588478) is planned to evaluate pirtobrutinib in combination with venetoclax versus obinutuzumab plus venetoclax, employing an MRD-guided approach (Table 1).

## 7. Pirtobrutinib Resistance and the Strategic Integration in CLL Management

Data from the BRUIN trials underscore the efficacy of pirtobrutinib in heavily pretreated patients with inclusion of those with dual-refractory disease, resistant to both cBTKis and venetoclax [30,31,37].

Despite these promising outcomes, a subset of patients ultimately develop acquired resistance to pirtobrutinib itself [30,31,37,47]. Post hoc analyses of the BRUIN cohort reveal that approximately 68% of patients experience disease progression accompanied by the emergence of mutations. Notably, BTK mutations are identified in 44% of cases, with PLCγ2 mutations present in 24%. Among these, mutations such as T474 and L528W are particularly prevalent, suggesting their significant role in resistance mechanisms [48].

Both T474I and L528W mutations are critical because they can lead to cross-resistance to multiple generations of BTK inhibitors, both cBTKi and ncBTKi [47,48]. The emergence of these mutations complicates treatment sequencing. Continued investigation is required to elucidate the precise mechanisms by which these mutations alter BTK function and to determine which BTK inhibitors retain efficacy in patients harboring these mutations [21,49,50]. Understanding these resistance pathways will inform the development of novel therapeutic strategies, including BTK degraders and T-cell engager therapies that can circumvent these mutations [26,50].

Clinically, the “mutator phenotype” characterizing these patients defines the propensity of CLL cells to acquire new genetic mutations that confer resistance to targeted therapies. These acquired mutations can drive treatment failure and disease progression, by activating parallel signaling pathways [21,48,49,50]. Additionally, they may contribute to the relatively short duration of responses observed with pirtobrutinib, with median remissions of approximately 8 months, particularly among double-refractory patients [37,48].

A recent post hoc analysis of the phase 1/2 BRUIN trial further elucidated the molecular landscape of pirtobrutinib response in patients with CLL previously treated with cBTKis [49]. The study focused on factors influencing sustained response versus disease progression and explored the potential for re-sensitization to cBTKis or other targeted therapies. Notably, early progressors, defined as those exhibiting disease progression within 24 cycles, more frequently had a history of cBTKi resistance and exhibited complex karyotypes. Patients harboring baseline BTK mutations, especially non-C481 variants, initially responded but eventually developed resistance, suggesting a mutator phenotype. Conversely, individuals with wild-type BTK or no detectable mutations experienced longer remissions, indicating that pirtobrutinib may have particular benefit in BTK inhibitor-naïve settings [49].

These data underscore the potential efficacy of pirtobrutinib, particularly in patients with double-refractory CLL, but they also suggest that a durable advantage will likely require integration of pirtobrutinib within a broader sequence that includes T-cell engager therapies [30,31,37,50]. A promising strategy is to employ pirtobrutinib to sustain disease control during a bridging period, with the aim of reducing tumor burden and mitigating related toxicities, and in selected patients, to switch to approaches capable of inducing deeper responses, such as CAR-T cell therapy [50]. Strategically incorporating pirtobrutinib into individualized treatment pathways with carefully timed transitions to cellular therapies may maximize the likelihood of long-term disease control and potential cure.

## 8. Improving BTKi-Safety with Pirtobrutinib

The interaction between BTK inhibitor therapy and individual patient comorbidities significantly influences toxicity profiles in CLL [12,51]. Many CLL patients are concurrently receiving antithrombotic therapy at treatment initiation, which elevates their risk for bleeding complications [12]. An analysis of bleeding events from the phase 1/2 BRUIN study (ClinicalTrials.gov NCT03740529), involving 773 patients treated with pirtobrutinib monotherapy, offers valuable insights into this interaction. Patients were stratified based on antithrombotic therapy exposure: 216 were antithrombotic therapy exposed (AT-E), and 557 were not (AT-NE). The AT-E group primarily received platelet inhibitors (51.9%), direct factor Xa inhibitors (36.6%), heparins (18.5%), salicylic acid derivatives (5.6%), and thrombolytics (2.3%), with warfarin contraindicated [52].

Bleeding or bruising events occurred in 44.9% of AT-E patients compared to 32.5% of AT-NE patients, predominantly within the first six months of therapy, with contusions being the most common. Grade ≥3 bleeding events were infrequent (2.8% in AT-E and 2.0% in AT-NE), with only one patient requiring dose reduction. These findings suggest that pirtobrutinib maintains an acceptable safety profile even in patients on concomitant antithrombotic agents [52]. Notably, pirtobrutinib is principally metabolized by the cytochrome P450 3A4 (CYP3A4) pathway [10]. Consequently, concomitant use of CYP3A4 inhibitors—such as certain anticoagulants and antiplatelet agents—can elevate pirtobrutinib exposure and may increase the risk of adverse effects. Therefore, patients receiving CYP3A4 inhibitors should be monitored for signs of bleeding, and dose adjustments of pirtobrutinib may be necessary [10].

Beyond bleeding, infection and cardiovascular risks remain central considerations [12]. Despite the challenges posed by the COVID-19 pandemic during the trial period, no patients discontinued therapy due to recurrent infections, indicating that pirtobrutinib’s infectious risk profile is favorable. This is particularly relevant considering the immunodeficiency inherent to CLL, affecting both humoral and cellular immunity [32,53]. Regarding cardiovascular safety, data demonstrated that most patients with prior cardiac events did not experience recurrence, and no discontinuations due to cardiac AEs occurred—an encouraging contrast to the discontinuation rates seen with other covalent BTK inhibitors [32].

These observations collectively support the hypothesis that pirtobrutinib offers a potentially improved safety profile relative to covalent BTK inhibitors, with lower incidences of cardiac, infectious, and bleeding toxicities [54].

To facilitate personalized treatment decisions, especially in elderly patients with complex comorbidities, we proposed the ‘CIRB’ criteria—a practical screening tool designed to evaluate key organ system vulnerabilities. The CIRB acronym encompasses Cardiovascular (C), Immunosuppression (I), Renal (R), and Bleeding (B) risk, including factors such as concomitant dual antiplatelet therapy [55]. Compared to existing tools like the Cumulative Illness Rating Scale (CIRS), the CIRB aims to more effectively capture the interplay between BTK inhibitor-related toxicities and patient-specific comorbidities, thereby guiding clinicians in selecting the most appropriate therapeutic strategy [55].

The phase 3 BRUIN CLL-314 trial (ClinicalTrials.gov NCT05254743) directly compares the efficacy and safety of pirtobrutinib with ibrutinib in treatment-naïve and previously treated CLL/SLL patients. The results of this head-to-head comparison with ibrutinib are anticipated and should be framed accordingly.

## 9. Conclusions

In conclusion, pirtobrutinib emerges as a potentially transformative agent within the evolving therapeutic landscape of chronic CLL, particularly for patient cohorts with limited treatment options due to intolerance or resistance to conventional covalent BTK inhibitors (Figure 2). Its favorable tolerability profile enables a safe administration in heavily pretreated populations, including those with dual resistance, underscoring its promise as an effective salvage therapy.

Resistance to pirtobrutinib often results from mutations in BTK, with certain mutations like A428D, T474I, and L528W potentially causing cross-resistance to other BTK inhibitors [27]. Although these patterns may influence future treatment sequencing, current understanding is limited, and optimal strategies for managing resistance are unclear [27]. In this setting, there is potential for BTK degraders that offer a distinct mechanism by removing BTK’s scaffolding role rather than inhibiting its enzymatic activity, differing from covalent and noncovalent BTK inhibitors [56,57]. Phase 1 data for NX-5948 and BGB-16673 show encouraging activity and manageable safety in heavily pretreated CLL [56]. T-cell engaging approaches, including bispecific antibodies, are also being explored; early CD3 × CD20 data for epcoritamab indicate favorable activity in relapsed disease [58].

Importantly, insights into the molecular determinants of response reveal that baseline genetic characteristics and prior treatment history significantly influence the durability of remission—patients with wild-type BTK tend to experience longer-lasting disease control, while those with complex genetic aberrations may encounter earlier progression [49]. This underscores the necessity of integrating pirtobrutinib within comprehensive, multimodal treatment strategies, such as bridging to advanced immunotherapies like CAR-T, to optimize outcomes and potentially enhance the efficacy of subsequent interventions [50]. By controlling the disease, pirtobrutinib prolongs survival and creates a window during which CAR-T cell therapy can be administered, a critical consideration for patients with limited treatment options. Disease stability reduces tumor burden and preserves the patient’s physical condition, thereby enhancing tolerance of the potentially intensive CAR-T-cell treatment. Lastly, bridging therapy aims to maintain a healthier and more stable clinical state prior to CAR-T infusion, a condition that supports the subsequent efficacy of CAR-T cells [50].

Preliminary safety considerations, including a manageable bleeding risk and a relatively low incidence of cardiovascular or infectious AEs, support pirtobrutinib’s clinical utility in CLL, a population typically characterized by advanced age and comorbidity [37,55]. Comparative trials will determine whether pirtobrutinib offers non-inferior efficacy relative to established covalent BTK inhibitors such as ibrutinib while maintaining a differentiated safety profile. As these data mature, pirtobrutinib is poised to assume roles in first-line and combination regimens. Ongoing investigations into resistance mechanisms and personalized treatment algorithms will be critical for extending remissions and moving toward long-term efficacy [50,59]. Ultimately, pirtobrutinib exemplifies how targeted, precision medicine approaches can significantly enhance disease management, improve patient quality of life, and shape the future of CLL therapy.

## Figures and Tables

**Figure 1 cancers-17-02974-f001:**
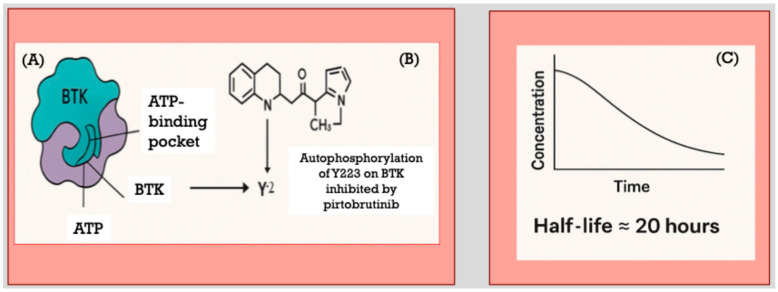
Unlike covalent BTK inhibitors, which irreversibly bind to the BTK active site, pirtobrutinib retains inhibitory activity even in the presence of mutations within this region, such as the Cys481 substitution (**A**). In addition, autophosphorylation of Y223 on BTK is potently inhibited by pirtobrutinib (**B**). Its favorable pharmacokinetic profile enables sustained BTK inhibition throughout the once-daily dosing interval (**C**).

**Figure 2 cancers-17-02974-f002:**
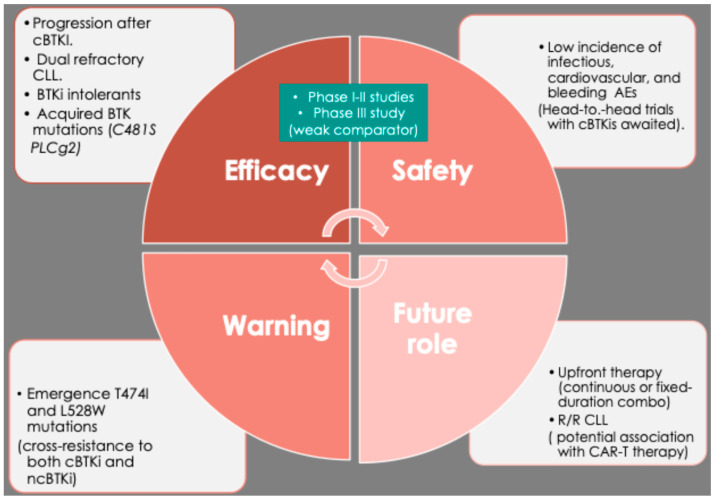
A graphical representation illustrating the efficacy, safety considerations, anticipated future role, and potential warnings associated with pirtobrutinib.

**Table 1 cancers-17-02974-t001:** Ongoing phase 3 clinical trials with pirtobrutinib.

Trial	Type of Study	Population	Experimental Arm	Control Arm
**NCT05023980**	Phase 3 (BRUIN CLL-313)	Untreated CLL/SLL	Pirtobrutinib	Bendamustine + Rituximab
**NCT04965493**	Phase 3 (BRUIN CLL-322)	Previously treated CLL/SLL	Pirtobrutinib + Venetoclax + Rituximab	Venetoclax + Rituximab
**NCT05536349**	Phase 2	Untreated CLL/Richter Transformation (RT)	Pirtobrutinib, Venetoclax, and Obinutuzumab	
**NCT06588478**	Phase 3 (CLL18)	Untreated CLL (MRD-guided approach)	Pirtobrutinib plus venetoclax	Venetoclax plus Obinutuzumab
**NCT05254743**	Phase 3 (BRUIN CLL-314)	Untreated CLL	Pirtobrutinib	Ibrutinib

## Data Availability

Data sharing is not applicable (only appropriate if no new data is generated or the article describes entirely theoretical research).

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
