# Peer review of "Pirtobrutinib in Chronic Lymphocytic Leukemia: Navigating Resistance and the Personalisation of BTK-Targeted Therapy"

_cancers, 2025, doi:10.3390/cancers17182974_

Round 1
Reviewer 1 Report
Comments and Suggestions for Authors
In this review paper the authors provides an overview of the clinical studies that have underpinned the incorporation of pirtobrutinib into the current therapeutic algorithm for CLL. They clarify the role pirtobrutinib in a first-line therapy and its potential integration with novel T-cell engager therapies for patients with relapsed/refractory CLL.
The paper References up to date and relevant and figures and tables are appropriate and well designed. .
Author Response
In this review paper the authors provides an overview of the clinical studies that have underpinned the incorporation of pirtobrutinib into the current therapeutic algorithm for CLL. They clarify the role pirtobrutinib in a first-line therapy and its potential integration with novel T-cell engager therapies for patients with relapsed/refractory CLL.
The paper References up to date and relevant and figures and tables are appropriate and well designed.
Response: We thank the referee for this favorable comment
In this review paper the authors provides an overview of the clinical studies that have underpinned the incorporation of pirtobrutinib into the current therapeutic algorithm for CLL. They clarify the role pirtobrutinib in a first-line therapy and its potential integration with novel T-cell engager therapies for patients with relapsed/refractory CLL.
The paper References up to date and relevant and figures and tables are appropriate and well designed.
Response: We thank the referee for this favorable comment
In this review paper the authors provides an overview of the clinical studies that have underpinned the incorporation of pirtobrutinib into the current therapeutic algorithm for CLL. They clarify the role pirtobrutinib in a first-line therapy and its potential integration with novel T-cell engager therapies for patients with relapsed/refractory CLL.
The paper References up to date and relevant and figures and tables are appropriate and well designed.
Response: We thank the referee for this favorable comment
Reviewer 2 Report
Comments and Suggestions for Authors
This high-quality narrative review on the use of pirtobrutinib in a spectrum of B-cell malignancies, notably Waldenström’s macroglobulinemia, diffuse large B-cell lymphoma, and Chronic Lymphocytic Leukemia (CLL) is of major importance and is of great interest to onco-hematology Centers and clinicians who treat patients affected by these diseases. Pirtobrutinib emerges as a potentially transformative agent within the evolving therapeutic landscape of CLL, particularly for patient cohorts with limited treatment options due to intolerance or resistance to conventional covalent Bruton’s tyrosine kinase (BTK) inhibitors. The data are presented clearly and comprehensively. They are also well-discussed. Literature citations are appropriate and balanced. The reviewer has only a few minor comments:
Point 1) It is customary in reviews to show the chemical structures of drugs mentioned in the text. In line, not all readers are familiar with the nomenclature and the chemical structures of drugs used for CLL treatment. Furthermore, one new Figure (Figure 1) is necessary to introduce the chemical structures of drugs used for CLL treatment, including the BTK inhibitors ibrutinib, acalabrutinib, zanubrutinib and pirtobrutinib, as well as venetoclax and bendamustine.
Point 2) Lines 174-175: Specify which anti-CD20 monoclonal antibodies (rituximab?, the glycoengineered anti-CD20 mAb obinutuzumab?, others?) and which type of chemotherapy are used. Line 235: briefly recall what is venetoclax.
Point 3) The two following reviews should be cited:
Shah and Stephens (in Case Reports Hematology Am Soc Hematol Educ Program 2021 Dec 10; 2021(1):68-75): the authors well summarized the clinical trials of BTK inhibitors and venetoclax that have investigated the role of anti-CD20 mAbs in frontline and relapsed settings of CLL treatment.
Nguyen et al (Scientific Reports volume 13, Article number: 9775, 2023, Efficacy and safety of add-on anti-CD20 monoclonal antibody to BTK inhibitor treatment for CLL: a meta-analysis): the authors conducted a systematic review and meta-analysis to compare the outcomes of combining anti-CD20 monoclonal antibodies with BTKi therapy versus BTKi monotherapy for CLL patients.
Author Response
This high-quality narrative review on the use of pirtobrutinib in a spectrum of B-cell malignancies, notably Waldenström’s macroglobulinemia, diffuse large B-cell lymphoma, and Chronic Lymphocytic Leukemia (CLL) is of major importance and is of great interest to onco-hematology Centers and clinicians who treat patients affected by these diseases. Pirtobrutinib emerges as a potentially transformative agent within the evolving therapeutic landscape of CLL, particularly for patient cohorts with limited treatment options due to intolerance or resistance to conventional covalent Bruton’s tyrosine kinase (BTK) inhibitors. The data are presented clearly and comprehensively. They are also well-discussed. Literature citations are appropriate and balanced.
Response: Thank you very much for your valuable comments and suggestions that have greatly contributed to enhancing the paper.
The reviewer has only a few minor comments:
Point 1) It is customary in reviews to show the chemical structures of drugs mentioned in the text. In line, not all readers are familiar with the nomenclature and the chemical structures of drugs used for CLL treatment. Furthermore, one new Figure (Figure 1) is necessary to introduce the chemical structures of drugs used for CLL treatment, including the BTK inhibitors ibrutinib, acalabrutinib, zanubrutinib and pirtobrutinib, as well as venetoclax and bendamustine.
- It is included in the context of Figure 1 showing that autophosphorylation of Y223 on BTK is potently inhibited by pirtobrutinib
Point 2) Lines 174-175: Specify which anti-CD20 monoclonal antibodies (rituximab?, the glycoengineered anti-CD20 mAb obinutuzumab?, others?) and which type of chemotherapy are used. Line 235: briefly recall what is venetoclax.
- This information was obtained from the seminal paper by Mato et al. (NEJM, 2023). Unfortunately, the information requested regarding the type of anti-CD20 antibody, Mo, and chemotherapy is not available in either the main text or the supplementary materials, as the data are presented in aggregate form.
Point 3) The two following reviews should be cited:
Shah and Stephens (in Case Reports Hematology Am Soc Hematol Educ Program 2021 Dec 10; 2021(1):68-75): the authors well summarized the clinical trials of BTK inhibitors and venetoclax that have investigated the role of anti-CD20 mAbs in frontline and relapsed settings of CLL treatment.
Nguyen et al (Scientific Reports volume 13, Article number: 9775, 2023, Efficacy and safety of add-on anti-CD20 monoclonal antibody to BTK inhibitor treatment for CLL: a meta-analysis): the authors conducted a systematic review and meta-analysis to compare the outcomes of combining anti-CD20 monoclonal antibodies with BTKi therapy versus BTKi monotherapy for CLL patients.
- Thank you for drawing attention on the papers mentioned below that are now added as reference.
- Shah HR, Stephens DM. Is there a role for anti-CD20 antibodies in CLL? Hematology Am Soc Hematol Educ Program. 2021 Dec 10;2021(1):68-75.
- Nguyen TT, Nhu NT, Tran VK, Viet-Nhi NK, Ho XD, Jhan MK, Chen YP, Lin CF. Efficacy and safety of add-on anti-CD20 monoclonal antibody to Bruton tyrosine kinase inhibitor treatment for chronic lymphocytic leukemia: a meta-analysis. Sci Rep. 2023 Jun 16;13(1):9775.
Reviewer 3 Report
Comments and Suggestions for Authors
The article by Stefano Molica and David Allsup is comprehensive and timely review manuscript on the role of pirtobrutinib in the treatment of chronic lymphocytic leukemia (CLL). The authors provide a thorough overview of the mechanism of action, clinical efficacy, safety profile, and emerging data on resistance mechanisms associated with this novel non-covalent BTK inhibitor. The topic is highly relevant given the growing population of patients who are intolerant or resistant to covalent BTK inhibitors (cBTKis). The manuscript is well-structured and cites extensive, up-to-date literature, including pivotal recent phase 3 data. The review is strong and informative but would benefit from significant revisions to improve clarity, depth in certain sections, and to ensure a balanced and critical perspective. The major areas for improvement are outlined below.
While the BRUIN trials (NCT03740529, CLL-321) are the cornerstone of pirtobrutinib's development, the review often reads like a summary of these trials rather than a critical synthesis of the available evidence. The tone at times veers towards promotional (e.g., "compelling efficacy," "transformative agent," "superior safety profile"). I suggest the authors to reframe the language to be more objective. For instance, instead of "superior safety profile," use "a differentiated safety profile" or "a potentially improved safety profile compared to cBTKis." Acknowledge the limitations of the data more explicitly (e.g., most data is from single-arm studies, cross-trial comparisons are indirect, longer follow-up is needed).
While discussing cross-resistance, the emergence of BTK mutations (T474I, L528W) that confer resistance to both covalent and non-covalent inhibitors is a critical clinical challenge. This should be discussed in more depth, including the implications for treatment sequencing.
While discussing efficacy in high-risk subgroups, the analyses from BRUIN CLL-321 are mentioned, the modest ORR (56%) in patients with PLCG2 mutations warrants a more nuanced discussion. What are the hypotheses for this? What does it imply for the mechanism of resistance?
While discussing comparative efficacy, the statement that ongoing trials "suggest that pirtobrutinib may offer non-inferior efficacy" (Page 9) is premature. The results of the head-to-head trial vs. ibrutinib (NCT05254743) are awaited and should be framed as such.
The note on Figure 2, "These are original figures generated using a Chat-GPT 5.0," is highly unorthodox and potentially concerning for scientific integrity. AI-generated figures must be meticulously checked for accuracy, and the prompt used must be disclosed if this is allowed by journal policy. It is preferable to use professionally designed original figures or adapted (with permission) published figures.
The reference list contains numerous formatting inconsistencies (e.g., journal names sometimes abbreviated, sometimes not; inconsistent use of "et al."; placement of year and DOI). Please ensure strict adherence to the journal's citation style guide.
Page 4, Paragraph 2: "many of whom had previously received covalent BTK inhibitors (with a median of four prior lines of therapy)" – This is a key point that highlights the heavily pre-treated population. It could be emphasized more strongly.
The update on the 317-patient cohort should explicitly state that this is a later analysis of the same BRUIN trial to avoid confusion.
The comparison of the IC arm (IdelaR/BR) should include a brief comment on why these were chosen as comparators (e.g., historical standards) and acknowledge that they are not considered optimal modern therapies, which may accentuate the benefit of pirtobrutinib.
The acronym "RT" is introduced but not used consistently. Please define it at first use (Richter Transformation) and use the acronym thereafter.
The sentence "Approximately 20% of patients... are typically older..." (Page 6) is vague. Cite a reference for this estimate.
Section 6 is a crucial section but feels brief. The concept of a "mutator phenotype" is introduced but not explained. A sentence elaborating on what this means in the context of CLL would be helpful for the reader.
The strategy of using pirtobrutinib as a "bridge" to CAR-T or other therapies is important. This could be expanded into a short paragraph discussing the practical considerations and goals of such a bridging strategy.
The conclusion is good but should be revised to more directly reflect the critical points raised in the review, including the promises and the current challenges (resistance, need for longer follow-up) associated with pirtobrutinib.
Typographical Errors:
Page 1: "Cancers 2025, 17, x" – Please update with the correct DOI once available.
Page 4: "He i salso local investigator" -> "He is also a local investigator"
Page 8: "prinviple" -> "principal"
Please perform a thorough proofread to catch other minor errors.
Author Response
The article by Stefano Molica and David Allsup is comprehensive and timely review manuscript on the role of pirtobrutinib in the treatment of chronic lymphocytic leukemia (CLL). The authors provide a thorough overview of the mechanism of action, clinical efficacy, safety profile, and emerging data on resistance mechanisms associated with this novel non-covalent BTK inhibitor. The topic is highly relevant given the growing population of patients who are intolerant or resistant to covalent BTK inhibitors (cBTKis). The manuscript is well-structured and cites extensive, up-to-date literature, including pivotal recent phase 3 data. The review is strong and informative but would benefit from significant revisions to improve clarity, depth in certain sections, and to ensure a balanced and critical perspective. The major areas for improvement are outlined below.
While the BRUIN trials (NCT03740529, CLL-321) are the cornerstone of pirtobrutinib's development, the review often reads like a summary of these trials rather than a critical synthesis of the available evidence. The tone at times veers towards promotional (e.g., "compelling efficacy," "transformative agent," "superior safety profile"). I suggest the authors to reframe the language to be more objective. For instance, instead of "superior safety profile," use "a differentiated safety profile" or "a potentially improved safety profile compared to cBTKis." Acknowledge the limitations of the data more explicitly (e.g., most data is from single-arm studies, cross-trial comparisons are indirect, longer follow-up is needed)
- Thank you for your comments. As you have suggested any attempt was done to temperate the tone for some too enthusiastic statemtnts. This is an example :” These findings establish that the third-generation BTK inhibitor pirtobrutinib possesses a proven efficacy and improved safety profile, with the promise to address several unmet needs associated with covalent BTK inhibitors.”
While discussing cross-resistance, the emergence of BTK mutations (T474I, L528W) that confer resistance to both covalent and non-covalent inhibitors is a critical clinical challenge. This should be discussed in more depth, including the implications for treatment sequencing.
- The point was in detail discussed: “Both T474I and L528W mutations are critical because they can lead to cross-resistance to multiple generations of BTK inhibitors, both cBTKi and ncBTKi . The emergence of these mutations complicates treatment sequencing. Continued investigation is required to elucidate the precise mechanisms by which these mutations alter BTK function and to determine which BTK inhibitors retain efficacy in patients harboring these mutations. Understanding these resistance pathways will inform the development of novel therapeutic strategies, including BTK degraders and T-cell–engaged approaches that can circumvent these mutations “
While discussing efficacy in high-risk subgroups, the analyses from BRUIN CLL-321 are mentioned, the modest ORR (56%) in patients with PLCG2 mutations warrants a more nuanced discussion. What are the hypotheses for this? What does it imply for the mechanism of resistance?
- We tried to clarify providing some hypothes: “There exists substantial heterogeneity among PLCG2 mutations observed in patients. Different mutations can vary in their functional impact on PLCγ2 activity, subcellular localization, and interaction with upstream (BTK, SYK) and downstream signaling components. Consequently, not all PLCG2 mutations yield identical phenotypic effects or drug responses; some variants may confer stronger resistance, while others produce modest or context-dependent changes. This heterogeneity implies that PLCG2 status cannot be treated as a single binary biomarker for resistance.”
While discussing comparative efficacy, the statement that ongoing trials "suggest that pirtobrutinib may offer non-inferior efficacy" (Page 9) is premature. The results of the head-to-head trial vs. ibrutinib (NCT05254743) are awaited and should be framed as such.
- Also this point is now clarified “The phase 3 BRUIN CLL-314 trial (ClinicalTrials.gov NCT05254743) directly compares the efficacy and safety of pirtobrutinib with ibrutinib in treatment-naïve and previously treated CLL/SLL patients. The results of this head-to-head comparison with ibrutinib are anticipated and should be framed accordingly.”
The note on Figure 2, "These are original figures generated using a Chat-GPT 5.0," is highly unorthodox and potentially concerning for scientific integrity. AI-generated figures must be meticulously checked for accuracy, and the prompt used must be disclosed if this is allowed by journal policy. It is preferable to use professionally designed original figures or adapted (with permission) published figures.
- Thank you. In this revised version original figures realized in ppt are included.
The reference list contains numerous formatting inconsistencies (e.g., journal names sometimes abbreviated, sometimes not; inconsistent use of "et al."; placement of year and DOI). Please ensure strict adherence to the journal's citation style guide.
- Appropriate changes were done according to Journal guidelines.
Page 4, Paragraph 2: "many of whom had previously received covalent BTK inhibitors (with a median of four prior lines of therapy)" – This is a key point that highlights the heavily pre-treated population. It could be emphasized more strongly.
This point is now underscored: “These results are particularly relevant given that the study enrolled a heavily pretreated population: patients had received a median of four prior lines of therapy, with many having prior exposure to covalent BTK inhibitors, venetoclax, and chemotherapy—with or without anti-CD20 antibody treatments—highlighting the challenging nature of this cohort.”
The update on the 317-patient cohort should explicitly state that this is a later analysis of the same BRUIN trial to avoid confusion.
It is now clarified :”An updated analysis of this trial,”
The comparison of the IC arm (IdelaR/BR) should include a brief comment on why these were chosen as comparators (e.g., historical standards) and acknowledge that they are not considered optimal modern therapies, which may accentuate the benefit of pirtobrutinib.
- This point is now clarifified: “Of note, the IdelaR/BR combination was selected as the comparator in the BRUIN CLL-321 trial because it represented a reasonable historical standard of care for relapsed/refractory CLL, particularly in patients previously treated with Bruton's tyrosine kinase inhibitors (BTKIs). However, neither IdelaR nor BR is considered an optimal modern therapy. Their inclusion highlights these limitations, thereby underscoring the significant clinical benefit and improved safety profile of pirtobrutinib”.
The acronym "RT" is introduced but not used consistently. Please define it at first use (Richter Transformation) and use the acronym thereafter.
- It was done
The sentence "Approximately 20% of patients... are typically older..." (Page 6) is vague. Cite a reference for this estimate.
- This part has been reworded: Additionally, in many patients with CLL, who are typically older and burdened with substantial comorbidities, RT may be amenable to palliative management; in this context, single-agent pirtobrutinib could constitute a reasonable therapeutic option [42].
Section 6 is a crucial section but feels brief. The concept of a "mutator phenotype" is introduced but not explained. A sentence elaborating on what this means in the context of CLL would be helpful for the reader.
It is now clarified: Clinically, the “mutator phenotype” characterizing these patients defines the pro-pensity of CLL cells to acquire new genetic mutations that confer resistance to targeted therapies.hese acquired mutations can drive treatment failure and disease progression, by activating parallel signaling pathways [21, 48-50]. Additionally, they may contribute to the relatively short duration of responses observed with pirtobrutinib, with median re-missions of approximately 8 months, particularly among double-refractory patients [37, 48].
The strategy of using pirtobrutinib as a "bridge" to CAR-T or other therapies is important. This could be expanded into a short paragraph discussing the practical considerations and goals of such a bridging strategy.
This is discussed more in detail “Importantly, insights into the molecular determinants of response reveal that baseline genetic characteristics and prior treatment history significantly influence the durability of remission—patients with wild-type BTK tend to experience longer-lasting disease control, while those with complex genetic aberrations may encounter earlier progression 49. This underscores the necessity of integrating pirtobrutinib within comprehensive, multimodal treatment strategies, such as bridging to advanced immunotherapies like CAR-T, to optimize outcomes and potentially enhance the efficacy of subsequent interventions 50. By controlling the disease, pirtobrutinib prolongs survival and creates a window during which CAR-T cell therapy can be administered, a critical consideration for patients with limited treatment options. Bridging therapy aims to maintain a healthier and more stable clinical state prior to CAR-T infusion, a condition that supports the subsequent efficacy of CAR-T cells. Disease stability reduces tumor burden and preserves the patient’s physical condition, thereby enhancing tolerance of the potentially intensive CAR-T–cell treatment 50.”
The conclusion is good but should be revised to more directly reflect the critical points raised in the review, including the promises and the current challenges (resistance, need for longer follow-up) associated with pirtobrutinib.
This part was implemented with inclusion of studies recently presented at 2024 ASH meeting abd dealing with BTK degraders and epcoritamab “Resistance to pirtobrutinib often results from mutations in BTK, with certain mutations like A428D, T474I, and L528W potentially causing cross-resistance to other BTK inhibitors 27. Although these patterns may influence future treatment sequencing, current understanding is limited, and optimal strategies for managing resistance are unclear 27. In this setting, there is potential for BTK degraders that offer a distinct mechanism by removing BTK’s scaffolding role rather than inhibiting its enzymatic activity, differing from covalent and noncovalent BTK inhibitors 56-57. Phase 1 data for NX-5948 and BGB-16673 show encouraging activity and manageable safety in heavily pretreated CLL 56. T-cell engaging approaches, including bispecific antibodies, are also being explored; early CD3×CD20 data for epcoritamab indicate favorable activity in relapsed disease 58.”
Typographical Errors:
Page 1: "Cancers 2025, 17, x" – Please update with the correct DOI once available.
Page 4: "He i salso local investigator" -> "He is also a local investigator"
Page 8: "prinviple" -> "principal"
Please perform a thorough proofread to catch other minor errors.
- It was done
Round 2
Reviewer 3 Report
Comments and Suggestions for Authors
No further comments